# Zearalenone Contamination in Corn, Corn Products, and Swine Feed in China in 2016–2018 as Assessed by Magnetic Bead Immunoassay

**DOI:** 10.3390/toxins11080451

**Published:** 2019-08-01

**Authors:** Ming Li, Chuqin Yang, Yuhao Mao, Xia Hong, Daolin Du

**Affiliations:** Institute of Environmental Health and Ecological Security, School of the Environment and Safety Engineering, Jiangsu University, Xuefu Road 301, Zhenjiang 212013, China

**Keywords:** zearalenone, contamination, magnetic bead immunoassay, corn, corn product, swine feed

## Abstract

In total, 405 samples of corn, corn products, and swine feed from China in 2016–2018 were surveyed for zearalenone (ZEN) contamination using a magnetic bead immunoassay-coupled biotin–streptavidin system (BAS-MBI). The developed BAS-MBI had a limit of detection (LOD) of 0.098 ng mL^−1^, with half-maximal inhibition concentration (IC_50_) of 0.71 ng mL^−1^ in working buffer, and an LOD of 0.98 ng g^−1^; the detection range was from 0.98 to 51.6 ng g^−1^ in authentic agricultural samples. The BAS-MBI has been demonstrated to be a powerful method for the rapid, sensitive, specific, and accurate determination of ZEN. The ZEN positivity rate reached the highest level of 40.6% in 133 samples in 2016; ZEN levels ranged from 1.8 to 1100.0 ng g^−1^, with an average level of 217.9 ng g^−1^. In 2017, the ZEN positivity rate was the lowest at 24.5% in 143 samples; ZEN levels ranged from 1.1 to 722.6 ng g^−1^, with an average of 166.7 ng g^−1^. In 2018, the ZEN positivity rate was 31.8% in 129 samples; ZEN levels ranged from 1.3 to 947.8 ng g^−1^, with an average of 157.0 ng g^−1^. About 20% of ZEN-positive samples exceeded maximum limit levels. An alternative method of ZEN detection and a valuable reference for ZEN contamination in corn and its related products in China are provided. This survey suggests the need for prevention of serious ZEN contamination, along with management for food safety and human health.

## 1. Introduction

Zearalenone (ZEN), which is a secondary metabolite from *Fusarium* species, has been frequently found in corn, corn product, feed, and cereal crops [1]. ZEN has become one of the most widespread contaminations of *Fusarium* toxins [2]. Neurotoxicity, teratogenesis, immune function, estrogenic, and even carcinogenic effects might be caused after consuming the products with ZEN contamination [3]. From the planting and harvesting of crops to the processing and storage of agricultural products, any part of the food chain might be contaminated by ZEN. The European Commission (EC) Regulation No. 1126/2007 regulated that the maximum limits (MLs) of ZEN should be no more than 100 ng g^−1^ in unprocessed cereals and 350 ng g^−1^ in unprocessed maize, respectively [4]. A provisional maximal tolerable daily intake for ZEN has been established as 0.5 ng g^−1^ of body weight by the Expert Committee of Joint FAO/WHO [5]. In China, the ML for ZEN has been regulated at 60 ng g^−1^ in wheat, wheat flour, corn, and corn flour [6].

Corn is a staple crop in agricultural regions, mainly for direct consumption or the processing and production of corn-related products around the world [7,8]. As corn is often contaminated with high levels of ZEN, corn-related agricultural products are inevitably contaminated by this mycotoxin (Zhang et al. 2018) [9]. Different MLs have been set by authorities, but the different contamination levels of ZEN may continue to occur with suitable space–time and environmental conditions. Thus, an integrated approach using ZEN occurrence rules, agricultural practices, environmental patterns, contamination levels, and risk assessments are critical for the prevention and control of ZEN contamination [10]. As a commonly occurring mycotoxin in agricultural products, studies on the analytical method of ZEN and the intensive monitoring of ZEN contamination should first and foremost guarantee the food safety and minimize the human health risks.

The current detection methods for ZEN mainly include HPLC [11], HPLC tandem mass spectrometry [12,13], and immunoassay [14,15]. Immunoassay has attracted the most attention for ZEN detection due to cost-effectiveness and high throughput screening capability [16,17,18,19]. A variety of immunoassays, such as enzyme-linked immunosorbent assay (ELISA) [20], chemiluminescence enzyme immunoassay (CLEIA) [21], fluorescence polarization immunoassay (FPIA) [22], colloidal gold-based lateral-flow immunoassay [23], and electrochemical immunoassay [24] have been introduced to detect ZEN. Then, the novel phage-displayed peptide and aptamer biomolecules were used to develop immunoassays for ZEN [25,26].

Magnetic bead immunoassay (MBI), a potential homogeneous assay which provides a fully reaction interface in magnetic particles as compared to the microtiter plate-based method, could be achieved with efficient magnetic separation and a convenient detection procedure [27,28]. The biotin–streptavidin system (BAS) is an ideal strategy to realize the amplification of signal and improvement of sensitivity [29]. With the help of the BAS, the labeling tracer amount of immunoassay can be significantly increased and the detection signal is more obvious, thus ensuring higher sensitivity and avoiding the use of a tracer-labeled second antibody [30]. Hence, a sensitive and highly efficient detection for harmful pollutants might be easily realized by combining the BAS system and MBI method.

In consideration of the common occurrence of ZEN and its harmful global implications, the contamination levels of ZEN in China are investigated in this study. For this purpose, a new immunoassay based on the separation technique of magnetism and signal amplification strategy of biotin–streptavidin (BAS-MBI) is developed and validated (Figure 1). Then, this sensitive and efficient immunoassay is applied in the determination of ZEN contamination levels in 405 samples of corn, corn products, and swine feed from 2016 to 2018 in China. Afterward, the ZEN contamination levels in these agricultural product samples are evaluated. The findings of this research could provide an alternative method for ZEN detection and could help to provide knowledge about the contamination situation of ZEN in China.

## 2. Results and Discussion

### 2.1. Optimum Condition of BAS-MBI

The key parameters were investigated to guarantee the ideal sensitivity and performance of BAS-MBI for detecting ZEN (Figure 2). The concentrations of magnetic antigen probe and biotinylated antibody probe were selected by the checkerboard method when the value of B_0_ reached 1.0 (Figure 2A). Under the optimal concentrations of antigen and antibody, the minimum concentration of streptavidin–horseradish peroxidase (HRP), which could ensure the signal amplification of BAS-MBI, was used in this study (Figure 2B). As shown in Figure 2C, an exiguity or excess of H_2_O_2_ could inhibit the value of B_0_, which is a disadvantage to the sensitivity of BAS-MBI. The effect of tetramethylbenzidine (TMB) concentration on the sensitivity is shown in Figure 2D. Finally, the concentrations of H_2_O_2_ and TMB in color-substrate buffer were 3 mmol L^−1^ and 0.4 mmol L^−1^, respectively. As shown in Figure 2E, the optimization of the incubation time for the competitive process of BAS-MBI was chosen as 30 min, with ideal detection sensitivity.

Under the criteria of a higher value of B_0_/ half-maximal inhibition concentration (IC_50_) and lower value of IC_50_, 0.8 ng mL^−1^ of magnetic antigen probe (1:8000 dilution), 0.15 ng mL^−1^ of biotinylated antibody probe (1:40,000 dilution), and 26.3 ng mL^−1^ of streptavidin–HRP (1:38,000 dilution) were selected as the optimal working concentrations for BAS-MBI (Table 1). Moreover, 5% methanol, 0.5 mol L^−1^ Na^+^, and a neutral pH in PBS buffer were chosen as the optimal working buffer for BAS-MBI.

### 2.2. Sensitivity

Under optimal conditions, the standard and calibration curves of ZEN by BAS-MBI were obtained by fitting the serial concentrations of ZEN standard and the values of B/B_0_ (%) (Figure 3). The developed BAS-MBI showed a limit of detection (LOD, IC_10_) of 0.098 ng mL^−1^, an IC_50_ of 0.71 ng mL^−1^, and a detection range (IC_10_–IC_90_) from 0.098 ng mL^−1^ to 5.16 ng mL^−1^ in the working buffer. According to the procedures of authentic sample pretreatment and extraction, the ZEN levels of samples were equivalent to a 10-fold dilution and the matrix effects were negligible. Thus, for the analysis of corn, corn products, and swine feed samples, the working range of BAS-MBI was from 0.98 ng g^−1^ to 51.6 ng g^−1^, with an LOD of 0.98 ng g^−1^.

According to the guidelines for the MLs of ZEN [4,5,6], the sensitivity of the developed BAS-MBI could be perfectly satisfied with the requirements for detecting ZEN. Compared with chromatographic methods which need a complex sample pretreatment process and expensive instruments, the BAS-MBI showed the advantages of simplicity, low-cost, and high-throughput screening of samples. Compared with ELISA, which needs five-step reactions and an original separation method, the BAS-MBI used three-step reactions in 1 h with simple magnetic separation, and the coating antigen step and second antibody step were reduced in the developed BAS-MBI. Thus, the developed BAS-MBI could obviously shorten the overall testing time and simplified analytical procedure.

### 2.3. Specificity

In the cross-reactivity (CR) study for the proposed BAS-MBI, the resulting low values of CRs toward ZEN analogs indicated the high specificity of alternative immunoassay (Figure 4). The CR values of ZEN for α-Zearalenol and β-Zearalenol were 12.0% and 9.5%, respectively. Meanwhile, the CR values for α-Zearalanol, β-Zearalanol, and Zearalanone were 0.50%, 0.45%, and 0.32%, respectively. When the concentrations of other mycotoxins (such as aflatoxin B_1_, deoxynivalenol, fumonisin B_1_, T-2 toxin, and ochratoxin A) were 1000 ng mL^−1^, the CRs values were lower than 0.07%. Therefore, the CRs of ZEN and its related compounds guaranteed the specific determination of ZEN using the developed BAS-MBI. It is worth noting that the CRs of ZEN derivatives (such as ZEN glucosides and ZEN sulfates) with respect to the proposed immunoassay were not evaluated and that possible CRs might occur in these derivatives.

### 2.4. Accuracy

The high accuracy of BAS-MBI has been demonstrated by the good correlations between the results of developed BAS-MBI and referenced HPLC (*Y* = 1.0272 *X* + 3.8267, *R*^2^ = 0.9683) (Figure 5) in authentic samples. This result suggested that the developed BAS-MBI could be a satisfactory method for simple, highly effective, sensitive, specific, and accurate detection of ZEN contamination in agricultural product samples.

### 2.5. Contamination Levels of ZEN

A total of 405 agricultural product samples were used to investigate the ZEN contamination levels from 2016 to 2018 in China. These samples were collected and detected by the developed BAS-MBI. The ZEN contamination levels are shown in Table 2. In 2016, the ZEN-positive samples were found in 54 samples (40.6% of 133 total samples), including 14 corn samples, 18 corn product samples, and 22 swine feed samples. Contamination levels ranged from 1.8 to 1100.0 ng g^−1^, with the average level of ZEN being 217.9 ng g^−1^. In 2017, the positivity rate of ZEN was 24.5% in 35 out of 143 samples, with values in corn, corn products, and swine feed samples being 11 out of 48, 11 out of 23, and 13 out of 72, respectively. ZEN concentrations ranged from 1.1 to 722.6 ng g^−1^, with an average level of 166.7 ng g^−1^. In 2018, 41 out of 129 samples (31.8%) were detected as being ZEN-positive. ZEN contamination levels ranged from 1.3 to 947.8 ng g^−1^, with an average level of 157.0 ng g^−1^, and the corn product samples had the highest detection rate of 54.3% in 19 out of 35 samples.

From an annual based ZEN contamination situation, the corn, corn product, and swine feed samples were heavily contaminated with ZEN, with the highest ZEN positive rate, the highest average contamination level, and the highest ZEN contamination level (1100.0 ng g^−1^ in a swine feed sample) in 2016. The ZEN-positive rate was lowest in 2017, and the lowest average ZEN level was found in 2018. In the survey years, corn, corn product, and swine feed samples all showed ZEN-positivity. As a whole, the lowest positivity rate was 18.0% in 72 swine feed samples in 2017, and the highest positivity rate was 75.9% in 29 swine feed samples in 2016. It was found that the plum rain season was relatively longer and the rain was relatively abundant in sample collection areas in 2016 as compared to other years. The significant increases of ZEN contamination levels may be a result of moist conditions, which may be conducive to the growth of *Fusarium* and the production of ZEN.

For the ZEN-positive samples in 2016–2018, the status of ZEN contamination at different levels is shown in Figure 6. According to the guidelines for MLs from European Commission and China, the contamination levels were classified into five grades: <60 ng g^−1^, 60–100 ng g^−1^, 100–350 ng g^−1^, 350–500 ng g^−1^, and >500 ng g^−1^ in positive samples. ZEN-positive samples had levels lower than 60 ng g^−1^ in more than 40.0% of positive samples, meeting the ML for China. At the same time, the contamination levels of 53.7% ZEN-positive samples in 2016, 54.3 samples in 2017, and 65.9% samples in 2018 were lower than the ML value of the European Commission in cereal, at less than 100 ng g^−1^. It is worth noting that 20.4% of samples in 2016, 17.2% of samples in 2017, and 19.5% of samples in 2018 had contamination levels greater than 350 ng g^−1^, exceeding the ML values of both the European Commission and China. Some samples even detected ZEN-positivity with contamination levels much higher than 500 ng g^−1^, reaching up to 1100.0 ng g^−1^. These observations indicated that the contamination levels of ZEN showed a high detection rate and were a commonly occurring problem in the investigated samples. Strict ZEN contamination detection, risk assessment, and effective controls would be of great value to ensure food safety and human health.

## 3. Conclusions

In summary, an MBI method based on BAS strategy was successfully established and applied to investigate the contamination levels of ZEN in 405 samples of corn, corn products, and swine feed in 2016, 2017, and 2018 throughout China. Then, the contamination levels and the distribution of ZEN were analyzed and evaluated. The developed BAS-MBI was found to be a simple, effective, sensitive, specific, and accurate form of ZEN screening technology. The ZEN in agricultural product samples in China had a high detection rate and high contamination levels. The ZEN detection rate was 40.6% in 133 samples in 2016, 24.5% in 143 samples in 2017, and 31.8% in 129 samples in 2018, with concentrations ranging from 1.1 to 1100.0 ng g^−1^. The highest ZEN concentrations were found in swine feed samples from 2016. The ZEN positivity rate was highest in 2016 and the lowest in 2017. The contamination levels of 20.4% of ZEN-positive samples in 2016, 17.2% of samples in 2017, and 19.5% of samples in 2018 exceeded the limits of the European Commission and China. This study suggested that ZEN contamination and risk in agricultural products in China was a serious issue. An alternative method of ZEN detection and the valuable information of ZEN contamination in China could be provided to the agricultural product monitoring institutions and government to better ensure food safety and human health.

## 4. Materials and Methods

### 4.1. Materials and Instruments

Surface carboxyl-functional group-modified magnetic-particles (300 nm, 50 mg/mL) were obtained from Wuxi BioMag Scientific Inc. (Wuxi, China). The horseradish peroxidase conjugated with streptavidin (streptavidin–HRP) and bovine serum albumin (BSA) were supplied by Nanjing GenScript Biotechnology Co., Ltd. (Nanjing, China). The ZEN antigen and monoclonal antibody against ZEN (ZEN-McAb) were prepared and stored in our laboratory. The standards for ZEN and its analogs were supplied by Sigma-Aldrich (St. Louis, MO, USA). The hydroxy-2,5-dioxopyrolidine-3-sulfonicacid sodium salt (sulfo-NHS), biotinyl-N-hydroxy-succinimide (BNHS), and 1-(3-dimethylaminopropyl)-3-ethylcarbodiimide hydrochloride (EDC) were purchased from Shanghai Aladdin-Reagent Co., Ltd. (Shanghai, China). Polyoxyethylene sorbitan monolaurate (Tween-20), methyl ethanesulfonate (MES), tetramethylbenzidine (TMB), and other chemicals were from Tansoole (Shanghai, China).

The MES buffer (50 mmol L^−1^ MES, pH 5.2, in H_2_O), CBS buffer (50 mmol L^−1^ carbonate-buffered saline buffer, pH 9.6), PBS buffer (10 mmol L^−1^ phosphate buffer saline, pH 7.4), PBST buffer (PBS buffer with 0.05% Tween-20), and color-substrate buffer (3 mmol L^−1^ H_2_O_2_ and 0.4 mmol L^−1^ TMB freshly prepared, in citrate buffer with pH 5.0) were used and prepared in our laboratory.

A magnetic separator was used to carry out the magnetic separation procedure (Tianjin, China). An Infinite M1000 PRO reader was used to the absorbance measurements at 450 nm (Tecan, Switzerland). A Heal Force centrifuge Neofuge 18 R was used for sample separation (Hongkong, China). The sonication was performed using a KQ2200 ultrasonic processor (Kunshan, China). The ultrapure H_2_O was used and purified by a Milli-Q water treatment system (Bedford, MA, USA). The HPLC series of Agilent 1260 coupled with a fluorescence detector (Wilmington, DE, USA) was used to verify the accuracy of developed immunoassay.

### 4.2. Preparation of Immunoprobe

To develop a highly effective and sensitive BAS-MBI for ZEN, two immunoprobes of magnetic antigen probe and biotinylated antibody probe were prepared. Firstly, the magnetic antigen probe was prepared through the coupling of magnetic-particles and ZEN-antigen by NHS/EDC activation method [29,31,32]. Well-dispered surface carboxyl-functional group-modified magnetic-particles (50 μL) were created with sulfo-NHS (19.8 mg in 200 μL H_2_O) and EDC (10.2 mg in 200 μL H_2_O) in MES buffer (550 μL) and were activated for 30 min at 25 °C. The activated magnetic-particles were separated in the magnetic field and washed twice by MES buffer. Then, the dialyzed ZEN-antigen (50 μL, 5.3 μg mL^−1^) was added into the activated magnetic-particles (in 950 μL MES buffer) and stirred for overnight at 4 °C. After separation and washing, the nonspecific sites of magnetic antigen probe were blocked by 2% BSA in PBS buffer and stored at 4 °C.

The biotinylated antibody probe was the key biochemical reagent to achieve signal amplification of the BAS system. The carbodiimide method was used to prepare the biotinylated antibody probe [33]. The ZEN-McAb were dialyzed against CBS buffer (100 mmol L^−1^, pH 9.2) for 4 h and adjusted to 2 mg mL^−1^. Then, the BNHS (7.6 mg in 200 μL DMSO) and ZEN-McAb were mixed at the quality ratio of 1:10, and incubated for 6 h at room temperature. After dialysis with PBS buffer for 12 h at 4 °C, the biotinylated antibody probe was dissolved in PBS buffer containing 3% BSA and 50% glycerol, and for cryopreservation.

### 4.3. Development and Evaluation of BAS-MBI

#### 4.3.1. Procedures of BAS-MBI

The serial concentrations of ZEN standard solutions (50 μL/tube, in PBS containing methanol), an optimal dilution of magnetic antigen probe (50 μL/tube, in PBS), and the biotinylated antibody probe (100 μL/tube, in PBS) were mixed in a tube and incubated for 30 min at 37 °C to perform the competitive process of immunoassay. Following the washing step of magnetic separation, the optimal dilution of streptavidin-HRP (100 μL/tube, in PBS) was added to proceed the step of BAS strategy for 10 min at 37 °C. After washing separation and adding the color-substrate buffer, the situation of immunoassay was revealed by the absorbance value.

#### 4.3.2. Optimization and Standard Curve

The concentrations of magnetic antigen probe, biotinylated antibody probe and streptavidin-HRP, the key factors of working buffer (contents of methanol, concentration of Na^+^ and pH value) were optimized to evaluate their influence on the sensitivity of the proposed BAS-MBI [34]. The concentrations of serial ZEN standard were plotted against the values of B/B_0_ (%) to obtain the standard curve (Software of Origin Program 7.0). The percentage of B/B_0_ means the absorbance value with ZEN (B) to the absorbance value absence ZEN (B_0_). The optimal parameters, which would show the highest ratio of B_0_/IC_50_ and the lowest of IC_50_, were selected to develop the most desirable BAS-MBI.

#### 4.3.3. Evaluation of BAS-MBI

The sensitivity of BAS-MBI was evaluated by the value of LOD, IC_50_, and detection range. The specificity of BAS-MBI was evaluated by the CR of ZEN toward the structural analogs of ZEN. The CR value was calculated according to the formula: CR (%) = 100 × IC_50_ for ZEN/IC_50_ for ZEN analog. The lower value of CR indicated that the specificity of immunoassay was higher.

The accuracy of BAS-MBI was evaluated by the correlation between the results of ZEN detection between the developed immunoassay and the referenced HPLC. For the referenced HPLC [10,35], the homogenized sample of corn, corn product or swine feed (2 g) was extracted with the extract solution of acetonitrile and water (5 mL, 9:1, *v*:*v*) for 1 min vortex blending, and then 30 min extraction under ultrasonic condition. After centrifugation at 10,000 r min^−1^ for 5 min, the organic phase was concentrated and the residue was cleaned through a ZEN immuno-affinity column (Huaan Magnech Bio-Tech Co., Ltd., Beijing, China). Then, the ZEN was eluted with methanol (0.5 mL) and filtered with a 0.22 μm membrane filter prior to the HPLC. The chromatographic column was Eclipse XDB-C18 (5 μm particle size, 150 mm × 4.6 mm). The detection conditions were as below: injection volume of 20 μL, column temperature of 30 °C, mobile phase of acetonitrile, water, and methanol (46:46:8, v/v/v), flow rate of 1.0 mL min^−1^, excitation wavelength of 274 nm, and emission wavelength of 440 nm.

### 4.4. Sample Collection, Pretreatment, and Detection

China is a largely agricultural country. Corn planting areas and corn yields already occupy more than 36 million hectares and provide 210 million tons of yield yearly, respectively [36]. Knowledge on contamination levels of ZEN in corn and its related products is important for food safety [37]. In this study, a total of 405 samples of corn, corn product, and swine feed were collected from March 2016 to December 2018 from different farms, feed companies, and agricultural product processing industries throughout China. In 2016, 133 samples were collected, including 56 corn samples, 48 corn product samples (such as corn flour, corn bran, corn germ meal, and corn starch), and 29 swine feed samples. In 2017, the total of 143 collected samples included 48 corn samples, 23 corn product samples, and 72 swine feed samples. In 2018, a total of 129 samples were collected, with samples of corn, corn product, and swine feed numbering 26, 35, and 68, respectively. The collected samples were completely mixed and finely ground. Finally, all samples were cryopreserved until analysis.

When detecting the ZEN contamination level, 1 g of homogenized sample was weighed into tubes. Then, 4 mL of mixed solution of methanol-PBS buffer (1:1, *v*:*v*) were used as extraction solution and added to dissolve the sample. After shaking extraction for 10 min with a vortex mixer, the mixture was centrifuged at 5000 r min^−1^ for 10 min. One milliliter of the supernatant solution was isolated and mixed with 1.5 mL PBS buffer, and then analyzed using the proposed BAS-MBI method. Each analysis was repeated three times.

## Figures and Tables

**Figure 1 toxins-11-00451-f001:**
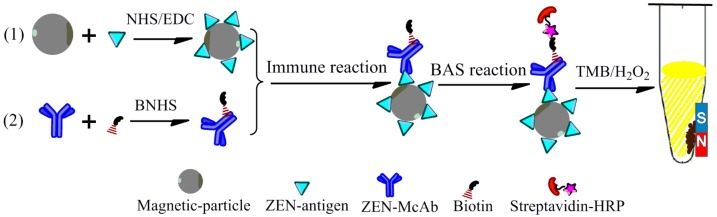
Schematic illustration of magnetism and signal amplification strategy of biotin–streptavidin (BAS-MBI) for zearalenone (ZEN). NHS: N-hydroxy-succinimide; BNHS: biotinyl-N-hydroxy-succinimide; EDC: 1-(3-dimethylaminopropyl)-3-ethylcarbodiimide hydrochloride; McAb: monoclonal antibody; TMB: tetramethylbenzidine; HRP: horseradish peroxidase.

**Figure 2 toxins-11-00451-f002:**
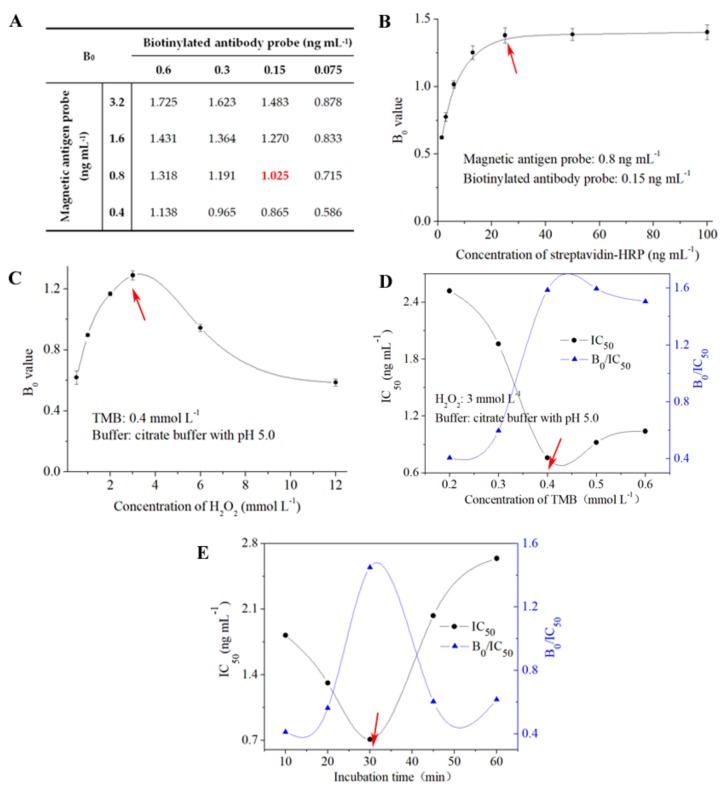
Optimization of the experimental parameters for BAS-MBI. (**A**) Concentration of antigen and antibody; (**B**) concentration of streptavidin–HRP; (**C**) concentration of H_2_O_2_; (**D**) concentration of TMB; (**E**) incubation time. B_0_: the absorbance value absence ZEN; IC_50_: half-maximal inhibition concentration.

**Figure 3 toxins-11-00451-f003:**
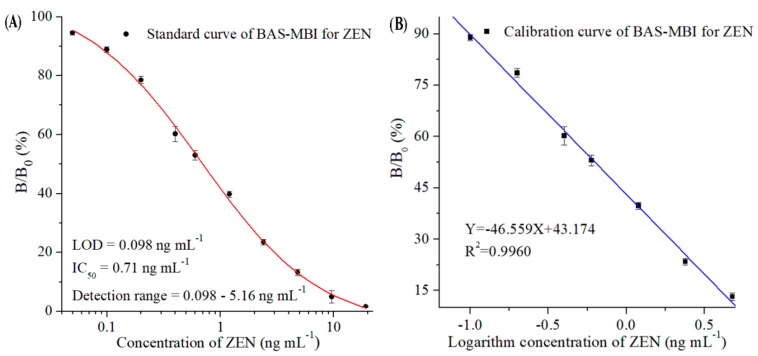
Standard curve (**A**) and calibration curve (**B**) of ZEN by BAS-MBI in the working buffer. B/B_0_ (%): the absorbance value with ZEN (B) to the absorbance value absence ZEN (B_0_); LOD: limit of detection.

**Figure 4 toxins-11-00451-f004:**
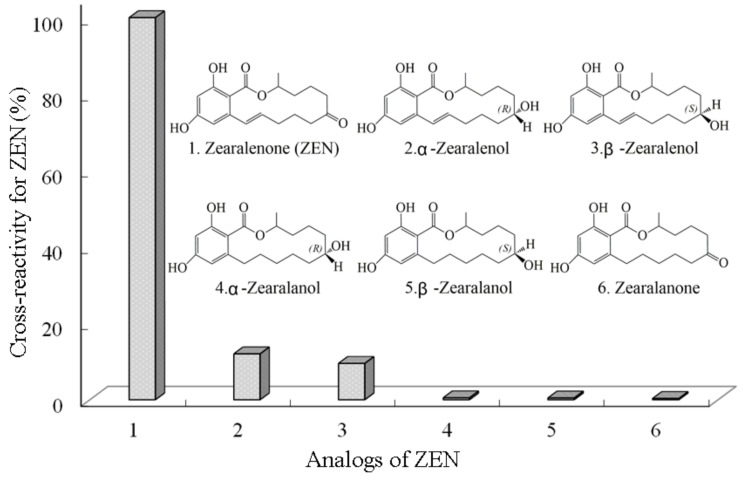
Cross-reactivities of ZEN toward its analogs by BAS-MBI.

**Figure 5 toxins-11-00451-f005:**
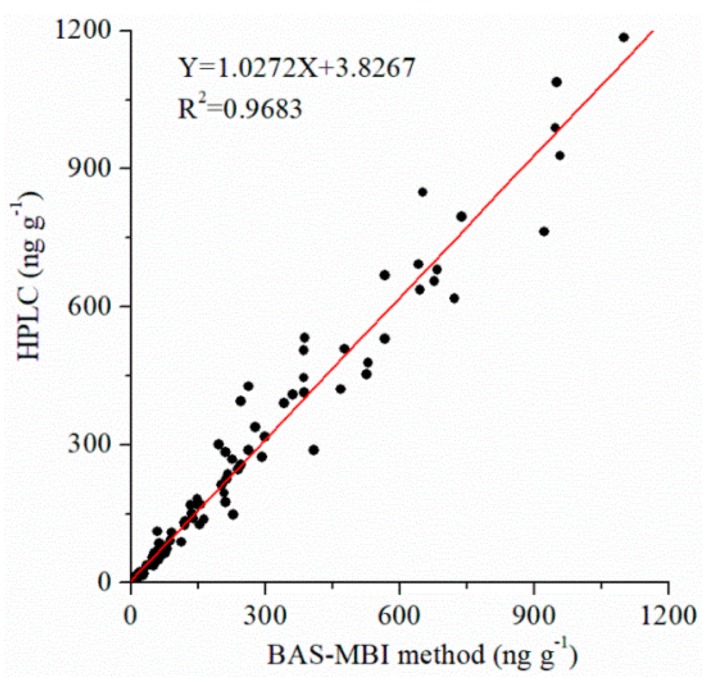
Correlation between the developed BAS-MBI and referenced HPLC for ZEN in authentic samples.

**Figure 6 toxins-11-00451-f006:**
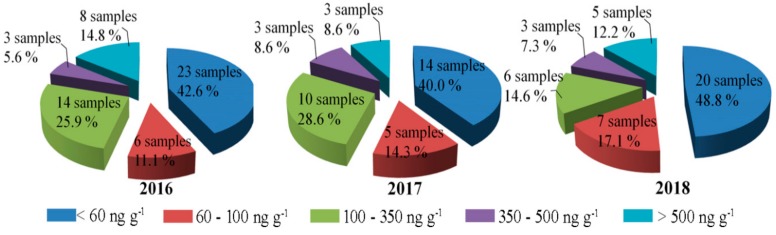
Distribution of ZEN-positive samples at different contamination levels in 2016–2018.

**Table 1 toxins-11-00451-t001:** Key parameters for the proposed BAS-MBI.

Factor	Parameter	Factor	Parameter
Magnetic antigen probe	1:8000 (0.8 ng mL^−1^)	Methanol (*v*/*v*, %)	5
Biotinylated antibody probe	1:40,000 (0.15 ng mL^−1^)	Na^+^ (mol L^−1^)	0.5
Streptavidin–HRP	1:38,000 (26.3 ng mL^−1^)	pH value	7.4

**Table 2 toxins-11-00451-t002:** ZEN-contamination distribution in 2016–2018 in China.

Year	Item	All	Corn	Corn Products	Swine Feed
2016	Total sample	133	56	48	29
	Positive sample	54	14	18	22
	Positive rate (%)	40.6	25.0	37.5	75.9
	ZEN range (ng g^−1^)	1.8–1100.0	1.8–950.4	2.3–958.1	8.8–1100.0
	Average ZEN (ng g^−1^)	217.9	144.1	315.0	185.4
2017	Total sample	143	48	23	72
	Positive sample	35	11	11	13
	Positive rate (%)	24.5	22.9	47.8	18.0
	ZEN range (ng g^−1^)	1.1–722.6	1.1–468.5	9.4–722.6	13.1–652.1
	Average ZEN (ng g^−1^)	166.7	87.2	205.3	201.3
2018	Total sample	129	26	35	68
	Positive sample	41	7	19	15
	Positive rate (%)	31.8	26.9	54.3	22.0
	ZEN range (ng g^−1^)	1.3–947.8	1.3–76.8	1.4–947.8	5.6–567.0
	Average ZEN (ng g^−1^)	157.0	24.1	176.7	193.9

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
