# Peer review of "Zearalenone Contamination in Corn, Corn Products, and Swine Feed in China in 2016–2018 as Assessed by Magnetic Bead Immunoassay"

_toxins, 2019, doi:10.3390/toxins11080451_

Round 1
Reviewer 1 Report
The manuscript is well structured, easy to understand, the authors clearly presented the problem, current developments and the aim of their study. The developed immunoassay is appropriate and well-illustrated for zearalenone determination. The manuscript might be accepted for publication after authors address several comments.
Chapter 2.1 Optimum condition of BAS-MBI
Additional data should be presented to support the optimization of key parameters.
1. Optimization of the mAb concentration - A dilution curve of the monoclonal antibody should be presented;
2. The effect of the ratio of the biotinylated antibody and Streptavidin-HRP on the BAS-MBI performance should be presented;
3. The effect of ionic strength – an ionic strength as high as 500 mM is questionable, usually over 100-150 mM can cause an increasing of IC50;
4. The optimization of the ratio between H2O2 and TMB substrate of HRP should be presented, since is well known that HRP can be inhibited by an excess of H2O2;
5. The optimization of the incubation time
Chapter 2.2. Sensitivity
The linear range of detection and the regression equation should be presented. How was LOD calculated? Which was the working range of concentrations?
Chapter 2.3. Specificity
What about cross-reactivity of this antibody to other mycotoxins such as deoxynivalenon, fumonisin B1, T2-toxin, ochratoxin A and aflatoxin B1?
Author Response
Dear reviewer:
We appreciate you very much for your professional comments and careful revision for our manuscript (Manuscript ID: toxins-558087). These comments and suggestions would help us in depth to improve the quality of our paper. We have revised the manuscript and answered the questions carefully according to your comments and submitted the files. The detailed corrections and responses are listed below item by item:
1. Optimization of the mAb concentration - A dilution curve of the monoclonal antibody should be presented.
Response: Thanks for your helpful suggestion. We have presented the optimization of the mAb concentration in our revised manuscript. The mAb concentration was in accord with the concentration of biotinylated antibody probe. To obtain the optimal mAb concentration, the checkerboard method was used to evaluate the ratio of the concentrations between biotinylated antibody probe and magnetic antigen probe. Under the criteria of the value of B0 reached 1.0, the optimal concentrations of mAb (biotinylated antibody probe) and magnetic antigen probe were 0.15 ng mL-1 and 0.8 ng mL-1, respectively. The result of optimization for the mAb concentration had been presented in Figure 2A and described in the revised manuscript. (Please show in Figure 2A and Line 85-87)
2. The effect of the ratio of the biotinylated antibody and Streptavidin-HRP on the BAS-MBI performance should be presented.
Response: Thanks for your helpful suggestion. We have presented the optimization of concentration of streptavidin-HRP. When the parameter of concentrations of antigen and antibody were fixed in the optimum condition, it had shown that the value of B0 could not keep growing with the increasing of the concentration of streptavidin-HRP. Finally, the minimum concentration of streptavidin-HRP, which could ensure the signal amplification of BAS-MBI, was used in this study. (Please show in Figure 2B and Line 87-89)
3. The effect of ionic strength-an ionic strength as high as 500 mM is questionable, usually over 100-150 mM can cause an increasing of IC50.
Response: Thanks for your comment and prompt. In our study, 10 mmol L-1 PBS buffer containing 0.5 mol L-1 Na+ was used as the diluent of biotinylated antibody probe. According to the procedures of BAS-MBI, the finally concentration of Na+ was 0.25 mol L-1 in the immune response system. At this condition, the higher value of B0/IC50 and lower value of IC50 for BAS-MBI had been found. Previous studies have shown that different antibodies have different tolerance to saline ions and other factors. Some studies have shown that the high salinity was favorable for improving the sensitivity of immunoassay. For example, Zhang et al. reported that the optimal Na+ was 0.32 mol L-1 in ELISA for metolcarb (Chin J Anal Chem, 2006, 34, 178-182). Cao et al. reported that the optimal Na+ was 0.5 mol L-1 for S-bioallethrin immunoassay (Anal. Methods, 2012, 4, 534-538). Zhang et al. reported that the optimal Na+ was 0.4 mol L-1 in immunoassay for aflatoxin B1 (Food Anal. Methods, 2018, 11, 2553-2560). Of course, there are many immunoassays with the lower optimal ionic strength. At your prompt, we will continue to focus on the effect of ionic strength on the sensitivity of immunoassays.
4. The optimization of the ratio between H2O2 and TMB substrate of HRP should be presented, since is well known that HRP can be inhibited by an excess of H2O2.
Response: Thanks for your helpful suggestion. The effect of concentrations of H2O2 and TMB in color-substrate buffer on the sensitivity and performance of BAS-MBI had been presented in our revised manuscript. It had been found that an exiguity or excess of H2O2 could inhibit the value of B0 and was to be disadvantage to the sensitivity of BAS-MBI. It could presume that an exiguity of H2O2 could not guarantee the fully catalytic reaction of HRP to TMB substrate. In addition, an excess of H2O2 might inhibit the activity of HRP. The effect of concentration of TMB on the sensitivity had been shown in Figure 2D. Finally, the concentrations of H2O2 and TMB in color-substrate buffer were used at 3 mmol L-1 and 0.4 mmol L-1, respectively. (Please show in Figure 2C, Figure 2D and Line 89-92)
5. The optimization of the incubation time.
Response: Thanks for your helpful suggestion. The optimization of the incubation time for the competitive process of BAS-MBI had been shown in Figure 2E and chosen as 30 min. Due to the higher value of B0/IC50 and lower value of IC50 for the BAS-MBI had been displayed at 30 min of incubation time. (Please show in Figure 2E and Line 92-94)
6. The linear range of detection and the regression equation should be presented. How was LOD calculated? Which was the working range of concentrations?
Response: Thanks for your valuable suggestion. The calibration curve of ZEN by BAS-MBI in the working buffer and the regression equation had been presented in Figure 3 in our revised manuscript, which could be able to show the linear range of detection. The value of LOD was calculated as IC10, which means 10 % inhibition concentration of ZEN for the immunoassay. The full name of LOD and detection range had been shown in the Section 2.2. Moreover, the working range was showed using the detection range (IC10–IC90), which range from 0.098 ng mL-1 to 5.16 ng mL-1 in the working buffer in our study. (Please show in Figure 3 and Section 2.2)
7. What about cross-reactivity of this antibody to other mycotoxins such as deoxynivalenon, fumonisin B1, T2-toxin, ochratoxin A and aflatoxin B1?
Response: Thanks. The specificity of the antibody to other mycotoxins (such as aflatoxin B1, deoxynivalenol, fumonisin B1, T-2 toxin and ochratoxin A) had been evaluated at 1000 ng mL-1 by the proposed immunoassay. The values of cross-reactivities (CRs) for these mycotoxins showed less than 0.07 %, thus indicated high specificity and guaranteed the specific determination of ZEN using the developed BAS-MBI. Moreover, this information had been added in our revised manuscript. (Please show in Page 4, Line 129-131)
Once again, thank you very much for your comments and suggestions. Here, we submit the revised manuscript and hope the revision will allow it proper enough to publish. There is any question, please inform us.
Best regards and looking forward to hearing from you.
Reviewer 2 Report
The authors describe the development of a magnetic bead immunoassay and its application to zearalenone (ZEN) analysis for a survey of food samples. The manuscript is well structured and written, citing the relevant literature. However, there are some issues specified below, which should be addressed in a revised version of the manuscript.
Line 33: Reference no. 4 should be replaced stating the relevant European Commission Regulation No. 1126/2007.
Line 50: It could be a little misleading when the authors say immunoassays have an advantage of specificity. Advantage over HPLC tandem mass spectrometry? HPLC-MS/MS is the method of choice for quantitative mycotoxin analysis because of its sensitivity, specificity, multi-analyte detection, and accuracy (trueness, precision) because of the possibility to use isotopic labelled standards. That should be stated very clearly. I agree with you regarding economic reasons and high throughput screening capability of immunoassays.
Line 106: Specificity is one of the most crucial properties when applying immunoassays. Therefore, I appreciate the conducted specificity experiments of the authors. However, the different zearalenol derivates are rather interesting when analyzing (reductive) ZEN metabolites. For corn and corn products it would be more relevant to test the cross-reactivity against masked/modified ZEN derivates such as ZEN-glucosides and ZEN-sulfates which could be contained in food samples with relevant concentrations. Thus, the authors are requested to point out that further ZEN derivatives (not tested in this study) could have possible cross-reactivities.
Author Response
Dear reviewer:
We appreciate you very much for your professional comments and careful revision for our manuscript (Manuscript ID: toxins-558087). These comments and suggestions would help us in depth to improve the quality of our paper. We have revised the manuscript and answered the questions carefully according to your comments and submitted the files. The detailed corrections and responses are listed below item by item:
1. Line 33: Reference no. 4 should be replaced stating the relevant European Commission Regulation No. 1126/2007.
Response: Thanks for your helpful suggestion. We have replaced the previous Reference by the European Commission Regulation No. 1126/2007 and described this regulation in our revised manuscript. (Please show in Reference 4 and Line 38-39)
2. Line 50: Advantage of specificity for immunoassays over HPLC tandem mass spectrometry? I agree with you regarding economic reasons and high throughput screening capability of immunoassays.
Response: Thanks for your helpful suggestion. I agree with your comments about the specificity of HPLC tandem mass spectrometry. And immunoassays have the characteristics of economy and high throughput screening capability. We have revised the characteristic description of immunoassays in our manuscript. (Please show in Page 2 and Line 56)
3. Line 106: For corn and corn products it would be more relevant to test the cross-reactivity against masked/modified ZEN derivates such as ZEN-glucosides and ZEN-sulfates which could be contained in food samples with relevant concentrations. Thus, the authors are requested to point out that further ZEN derivatives (not tested in this study) could have possible cross-reactivities.
Response: This is a very valuable suggestion. We have made a statement that the cross-reactivities of ZEN derivatives (such as ZEN-glucosides and ZEN-sulfates) toward the proposed immunoassay were not evaluated and the possible cross-reactivities might occur in these derivatives in Section 2.3. (Please show in Page 5, Line 133-135)
Once again, thank you very much for your comments and suggestions. Here, we submit the revised manuscript and hope the revision will allow it proper enough to publish. There is any question, please inform us.
Best regards and looking forward to hearing from you.
Round 2
Reviewer 1 Report
Satisfactory revisions have been made.